# Do Girls Have an Advantage Compared to Boys When Their Motor Skills Are Tested Using the Movement Assessment Battery for Children, 2nd Edition?

**DOI:** 10.3390/children10071159

**Published:** 2023-07-01

**Authors:** Bouwien Smits-Engelsman, Dané Coetzee, Ludvík Valtr, Evi Verbecque

**Affiliations:** 1Physical Activity, Sport and Recreation, Faculty of Health Sciences, North-West University, Potchefstroom 2520, South Africa; dane.coetzee@nwu.ac.za; 2Department of Health & Rehabilitation Sciences, Faculty of Health Sciences, Cape Town University, Cape Town 7701, South Africa; 3Department of Natural Sciences in Kinanthropology, Faculty of Physical Culture, Palacký University Olomouc, 771 11 Olomouc, Czech Republic; ludvik.valtr@upol.cz; 4Rehabilitation Research Centre (REVAL), Faculty of Rehabilitation Sciences, Hasselt University, Agoralaan Building A, 3590 Diepenbeek, Belgium; evi.verbecque@uhasselt.be

**Keywords:** motor competence, sex differences, validity

## Abstract

This study aims to investigate sex-related differences in raw item scores on the Movement Assessment Battery for Children, 2nd Edition (MABC-2) in a large data set collected in different regions across the world, seeking to unravel whether there is an interaction effect between sex and the origin of the sample (European versus African). In this retrospective study, a secondary analysis was performed on anonymized data of 7654 children with a mean age of 8.6 (range 3 to 16; SD: 3.4), 50.0% of whom were boys. Since country-specific norms were not available for all samples, the raw scores per age band (AB) were used for analysis. Our results clearly show that in all age bands sex-related differences are present. In AB1 and AB2, girls score better on most manual dexterity and balance items, but not aiming and catching items, whereas in AB3 the differences seem to diminish. Especially in the European sample, girls outperform boys in manual dexterity and balance items, whereas in the African sample these differences are less marked. In conclusion, separate norms for boys and girls are needed in addition to separate norms for geographical regions.

## 1. Introduction

The Movement Assessment Battery for Children, Second Edition (MABC-2) is a norm-referenced measurement tool designed to assess motor competence in children between the ages of three and sixteen, inclusive [1,2]. More specifically, the primary function of the test is to help identify children “at risk of” or presenting with a definite motor impairment [1,2]. Since the test covers a large age range, it comprises three age bands (age 3–6, age 7–10 and age 11–16). Within each age band a range of both gross and fine age-specific motor skills are assessed. These are grouped under three headings: manual dexterity (fine motor, three items), aiming and catching (gross motor, two items) and balance (gross motor, three items) [1]. The test is used in many countries around the world for many different purposes and is recommended in the international guidelines on Developmental Coordination Disorder (DCD) as one of the tests for use as part of the diagnostic process [3]. 

The development of any assessment instrument is an ongoing process, with updating of norms recommended every 10 to 15 years. Once a test has been published, further development takes many forms. For example, there will always be aspects of the psychometric properties of a test relating to reliability and validity that can be further explored. A good example of this is the question of test-retest reliability, which is often only evaluated over a small range of time intervals. Similarly, assessing validity is not an all-or-nothing process. Whereas the recommended process for determining predictive validity is straightforward, within the context of construct validity, the question of whether comparisons between different cultures are appropriate is less clear. Since it is generally accepted that no test, whatever the domain of behavior, is universally applicable, local norms are the ideal. However, the production of norms is an expensive process, so small cross-cultural studies are usually undertaken to locate any substantial differences between the population making up the original standardization and the new population of interest. If the differences are too large to be accommodated, then full standardization may follow. However, often these studies are not hypothesis-driven with specific predictions being made regarding the direction or extent of any potential differences. 

Since its publication in 2007, there have been numerous studies involving the MABC-2. These included a recent systematic review [2] which indicated that, overall, the test has good reliability between testers, within testers and between sessions. With regard to validity, good content and predictive validity have been reported, but there are other areas where less consistent outcomes are evident. Of particular relevance to the present study are investigations of construct validity as it applies to the suitability of the test for different cultures. As noted above, the question of whether any test can be truly “universal” is debatable. Instead, what is of more practical relevance is to document the nature of the differences between cultures so that adaptations to the test can be made where necessary and/or new norms produced. As an example, the research team led by Smits-Engelsman undertook a number of studies identifying differences between children in the UK, the Netherlands and Belgium, which then led to the collection of region-specific normative data developed for the Netherlands and Flanders (Belgium) [4]. Similar studies in China [5], Brazil [6], the Czech Republic [7,8] and Spain [9] then followed. In contrast, differences reported in Italy [10], and Japan [11,12] have not been followed up by the collection of local norms. 

Another aspect of construct validity that has not received enough attention is hypothesis testing. In particular, the impact of sex on the presentation of the MABC-2 test scores has not been sufficiently investigated; the norms were published for both sexes combined in both the 1992 edition of the Movement ABC manual and its revision in 2007 [1,4,5,6]. This is surprising, since it is well established that boys outperform girls on object control tasks [13,14,15,16,17,18] and girls are better at locomotor [16,19,20] and balance tasks [13,14,15]. These differences are usually attributed to how they spend their leisure time: boys usually play more with balls, while girls do activities such as rope-skipping and other balancing tasks related to gymnastics [21]. 

Several standardized motor tests do acknowledge sex differences—as mentioned above—through their separate norms for boys and girls, e.g., the Bruininks-Oseretsky Test of Motor Proficiency, 2nd Edition (BOT-2; boys better at strength, running speed and agility tasks) [22], the Test of Gross Motor Development, 2nd and 3rd Editions (boys better at object control and girls at locomotor tasks [23]), and the Körperkoordinationstest für Kinder 3+ (boys better at side jumping and eye-hand coordination, girls better at backward walking) [21]. However, there are few studies that have explored differences in MABC performance between boys and girls. It is striking to note that in the literature, when samples of children with DCD are recruited, the prevalence of boys is much higher when the MABC-2 is used (i.e., 68.8% (±12.5) boys [24,25,26,27,28,29,30,31,32,33,34,35]) compared to the BOT-2 (i.e., 48.4% (±17.1) boys [36,37,38]). This leads to the assumption that sex-related differences can impact the normative data and thereby potentially cause the underidentification of girls with DCD, which in turn reflects on the test’s construct validity.

As shown in Table 1, important sex differences seem to be present in normative samples as well [7,11,12,19,39,40,41,42]. In the majority of the studies, girls performed better on the balance scale [7,11,12,19,40,41,42] and on manual dexterity [7,12,19,39,40], whereas contrarily to what would be expected [14], boys are only sometimes superior in aiming and catching [40,41,42]. It does not come as a surprise that girls’ total score is also often better [7,11,12,19,39,40]. Sex-related differences in a group of healthy children may be explained by the onset of puberty, which occurs earlier in girls compared to boys [43], as it is known to trigger improvements in neurological, muscular, skeletal, and endocrine systems [44]. The differences shown in Table 1, however, mainly concern preschoolers (age 3–6), whereas early onset of puberty occurs well beyond this age [44]. This indicates that other environmental factors may be in play, such as family, school context and cultural expectations (religion and local sports), that may be different for boys and girls [45]. For instance, even though, worldwide, boys are better at ball skills, even as young as 15–23 months old [46], Aboriginal girls throw better than children from other cultures, which can be explained by their cultural belief that throwing for hunting and defense is important for both sexes [47]. Another aspect that has been suggested to impact motor skill development is socioeconomic status (SES). Several authors indicate that low SES has a negative impact on motor skills in children [20,48,49], whereas others do not report differences [50]. Whether or not SES has a negative impact on skills seems to depend upon the region and the type of skills being assessed [13,51,52,53], but the interaction effect with sex remains under debate [49,53].

This study therefore aims to investigate sex-related differences in raw MABC-2 item scores in a large data set collected in different regions across the world. As such, we seek to unravel whether there is an interaction effect between sex and the origin of the sample (European versus African). Answers will be sought to the following research questions: Are there sex-related differences in raw MABC-2 item scores?How do children on different continents perform, and are there sex-related differences in raw MABC-2 item scores on different continents?Are the sex-related differences in raw MABC-2 item scores age-dependent?

Since the domain scores are the scaled sums of the standardized item scores, sex-related differences at an item level should be explored first. We hypothesize that, overall, girls will outperform boys on the manual dexterity and balance items, and that boys will be better at aiming and catching than girls. Because of the daily activities of the children and their sports participation, we expect that these differences will be clearly present in the European sample.

## 2. Materials and Methods

### 2.1. Participants

In this retrospective study, a secondary analysis was performed on anonymized data collected during several previous projects [4,54,55]. The registered data were delivered anonymized by Pearson and the co-authors to the first author (BSE) and could not be linked to the participants in any of the countries. The sample consisted of 7654 children with a mean age of 8.6 (range 3 to 16; SD: 3.4), 50.0% of whom were boys. Some of the participants formed parts of carefully stratified samples of children involved in country-specific test standardizations, whereas other participants were randomly chosen from larger samples to explore test validity. 

The characteristics of the entire sample are summarized in Table 2. As the table shows, most children (75.2%) were European (mean (SD) age: 8.1 (3.5)). Half of this group came from the Netherlands (50.5%), 20.1% from the UK, 19.2% from Belgium and 10.2% from Czechia. The remaining 25% of the children live in Africa and in this sample the youngest age group (3–5) was missing, making this African sample significantly older (24.8%, mean (SD) age: 10.0 (2.8)). The African subsample consisted of children from South Africa (70.9%), Ghana (20.6%) and Nigeria (8.5%). Both sexes were equally represented in all subsamples (Table 2). The distribution across the age bands within each subsample is depicted in Table 2. 

### 2.2. Movement Assessment Battery for Children, Second Edition

The original and Dutch versions of the MABC-2 were used in the present study [1,56]. In content, the UK and Dutch versions are identical. The test consists of three age bands (AB1 for 3- to 6-year-olds, AB2 for 7- to 10-year-olds, and AB3 for 11- to 16-year-old children), for which eight age-specific items have been defined as representative of three domains: manual dexterity (MD; 3 items), aiming and catching (A and C; 2 items) and balance (B; 3 items). In all cases, test administration followed the standardized procedure in the manual. Since country-specific norms were not available for all samples, for the purpose of this study, the raw scores were used for analysis.

### 2.3. Statistical Analysis

Statistical analyses were performed with SPSS 28.0 for Windows. The sample was described using demographic data (age, sex) and the distribution across the MABC-2 age bands and continents (Europe versus Africa) from which the children were recruited. The Shapiro–Wilk test was used to check for normality. The data for the total sample (Figure 1) and for the European and African subsamples were extremely skewed for items 2, 3, 4, 6, 7 and 8. 

To compare the children’s age between the subsamples (European versus African), an independent Student’s *t*-test was applied. To compare the sex distribution and age-band distribution across the subsamples, a Chi-squared test was used. Subsample differences (boys versus girls or European versus African subsamples) within each age band were explored with the Mann–Whitney U test. Subsets were composed for each age band to explore sex differences within each subsample. For analyses comparing performances between European and African subsets, only the 6-year-old children were selected, as the AB1 African subsample consisted exclusively of children aged 6 years old. 

## 3. Results

### 3.1. Age Band 1

Sex differences were present for all items, except for posting coins with the preferred (*p* = 0.374) and non-preferred (*p* = 0.627) hand (Table 3). The girls were better at threading beads (*p* < 0.001), drawing a trail (*p* < 0.001), standing on one leg on the preferred and non-preferred leg (*p* < 0.001), walking with heels raised (*p* < 0.001) and jumping on mats (*p* < 0.001) than the boys. The boys outperformed the girls on catching (*p* = 0.025) and throwing a bean bag (*p* < 0.001). Table 4 shows the differences between the sexes within each subsample. Across all subsamples, girls were better at drawing a trail and jumping on mats. The other items differed depending on the subgroup. The girls usually outperformed the boys in the European subsample, except for the aiming and catching items, where they performed similarly. In the African sample, no differences were found between the sexes for most items, except for drawing a trail (girls better than boys), throwing a bean bag (boys better than girls) and jumping on mats (girls better than boys). Details on item performances for the subsamples are provided in Table A1.

### 3.2. Age Band 2

As shown in Table 3, boys and girls performed significantly differently on all items (*p* < 0.001). The girls were better than the boys at all manual dexterity and balance items (*p* < 0.001). The boys outperformed the girls on the aiming and catching items (*p* < 0.001). Within the European subsample identical results were found to those for the entire group (Table 4). In the African subsample the results deviated from the entire group, as there were no differences between boys and girls for placing pegs with the non-preferred hand (*p* = 0.186), drawing a trail (*p* = 0.426) and walking heel-to-toe forward (*p* = 0.711) (Table 4). Details for the differences between the European and African subsamples are provided in Appendix A.

### 3.3. Age Band 3

In this age band the sex differences were less marked (Table 3). The girls were better at turning pegs with their preferred hand (*p* < 0.001) and drawing a trail (*p* < 0.001). The boys outperformed the girls on catching a ball with their preferred and non-preferred hand (*p* < 0.001), throwing a ball at a wall-mounted target (*p* < 0.001) and walking heel-to-toe backwards (*p* = 0.013). In the subsamples, manual dexterity and aiming and catching were similar in the European sample to the entire group, whereas drawing a trail was not significantly different between boys and girls in the African subsample (*p* = 0.663). For the balance subscale, the sex differences between the European and African subsamples were most divergent. In the European subsample, girls outperformed boys in the two-board balance task (*p* = 0.023) and in zig-zag hopping on the non-preferred leg (*p* = 0.044), whereas in the African subsample, boys were better than girls at the two-board balance task (*p* = 0.031) and walking heel-to-toe backwards (*p* < 0.001). 

## 4. Discussion

The aim of this study was to unravel whether there is an interaction effect between sex and the origin of the sample. As hypothesized, our results clearly show that in all age bands, sex-related differences are present. In AB1 and AB2, girls are superior on most items, with the exception of A and C, whereas in AB3 the differences seem to diminish. Within the subsamples, these differences are not as straightforward, but are still present. Especially in the European sample, girls outperform boys in manual dexterity and balance items, whereas in the African sample these differences are less marked.

Since the international guidelines on DCD recommend the test’s use in the diagnostic process [3], sound normative data are a cornerstone, as the result plays a crucial role in determining whether or not a child receives additional support (e.g., at school) or even treatment (e.g., physiotherapy or occupational therapy). However, our results clearly show that the normative raw data are extremely skewed (Figure 1), indicating that children either can or cannot perform the tasks, so the use of standard scores following a bell-shaped distribution is highly questionable. For example, one mistake less or more on the drawing trail can have a huge impact on the meaning of the result. This lack of distribution in the data not only adversely affects the diagnostic accuracy of a test, but also impacts the test’s ability to detect changes. Furthermore, similar to what has been reported in the literature in the form of domain and total scores (summarized in Table 1) [7,11,12,19,39,40], our results reveal a marked difference at the item level in raw scores between boys and girls, indicating that separate sex-specific norms are imperative. When boys systematically perform worse compared to girls and combined normative data are applied, the chance of boys being identified as performing below the norm is higher compared to girls, with a potential risk of false positive identification of motor-skill deficits. On the other hand, girls are at risk of remaining unidentified, especially when the motor difficulties are subtler. If so many items favor girls, total scores will also be misleading, since no corrections for sex differences have been implemented so far. Either the items in the MABC-2 are less suited for boys or too much linked to skills in which girls tend to be superior, which directly reflects the test’s content validity. 

This does raise questions about the test’s composition and, therefore, the item choices, besides building on earlier versions of the test (TOMI). For example, why was it decided that jumping in a square, as in hopscotch, is a more important task to include than jumping over a ditch (as in a long jump) or jumping towards a hoop (closer to a vertical jump); why did we choose threading beads and not pressing phone keys or building a tower of small blocks; why was aiming chosen and not throwing; and why is there no item intended to measure agility or items close to having the skills needed for personal care? These choices have very important consequences, since the types of tasks are prone to sex-related differences [13,14,57,58,59].

There seems to be a consensus about what comprises “fundamental motor skills,” and that there are three categories of items that need to be included (locomotor, ball skills and balance). For example, regardless of their age, boys usually perform better on the object control skills in the second edition of the Test of Gross Motor Development (TGMD-2) [13,14], and girls perform better on the TGMD-2’s locomotor skills [14]. When children are assessed using the Athletic Skills Track (AST), where they have to balance, hopscotch, do traveling jumps, slalom, roll, run, alligator crawl and clamber as quickly as possible, requiring good physical fitness, boys outperform girls, regardless of age [60,61]. When children are asked to maintain a posture for a predefined time period (e.g., standing on one leg for 30 s), girls tend to outperform boys, whereas boys seem to do better in more dynamic situations such as balance during walking or performing reaching tasks [59]. Interestingly, most items of the MABC-2 emphasize accuracy and precision, which require both motor control and sustained attention, or enforces an accuracy-speed tradeoff for the manual dexterity items. Girls are more likely to have superior manual control abilities for performing novel tasks [62] and overall better inhibition control, which increases their selective attention, whereas boys are usually faster [63]. As such, the MABC-2 items seem easier for girls to perform, as they tap into their strengths. This also raises the question of whether a comparable percentage of boys would be diagnosed with DCD if there were more emphasis on gross motor skills and actual performance rather than accuracy (e.g., running, picking up an object and sprinting back, or a high or broad jump instead of an accuracy jump), or whether there would be a higher prevalence of DCD girls instead of boys.

One of the key environmental factors in the emergence of such differences is the sex stereotypes or gender role models that influence motor development from toddlerhood onwards [58]. Girls and boys are often encouraged to practice different types of sports, spend their leisure time differently, and perform different types of activities [45]. A first step towards a more accurate formal assessment using the MABC-2 would be to establish sex-specific norms. Yet, the sex differences in our sample were quite distinct depending on the child’s background (European versus African), which emphasizes the important role of the environment. It seems that the emphasis in European culture is more on fine motor play for girls, such as coloring and fine motor games, which may increase the (natural?) differences between genders. Hence, to apply motor tests in an environmentally valid way, it would make more sense to either develop contemporary regional gender-specific MABC-2 norms, or to incorporate tasks that are closer to children’s actual daily activities.

Our results make clear that when we state that a child has “poor motor skills,” this assumption depends upon the items in the test (some having items that boys excel in and others having more items that favor girls) and secondly on the samples used for the norms. Motor competence is defined as the ability to perform a wide range of motor skills. However, what should be in this “wide range” is less obvious. In Europe, hitting a ball with a baseball bat or eating with chopsticks would not be seen as culturally appropriate test items to evaluate motor skills, as they do not reflect children’s daily activities. On the other hand, how many boys spend time playing with beads or ministacks? Items in the MABC-2 were chosen to be as culturally independent as possible, which to a large extent was successful, by avoiding sports-related skills (jumping or throwing for distance). Given the fact that recent training paradigms are task-oriented and focus on the identification of activities children struggle with, standardized tests should also consider implementing those elements that are relevant to a child’s wide range of daily tasks. For instance, tasks that require running fast without falling over or stepping on an object can be considered motor skills that belong in this wide range. Given the decreasing level of daily physical activity, such items should get a higher priority and should be integrated into future norms.

Norm-referenced tests are calibrated carefully on the representative sample for which the norms are intended, which was done optimally for the UK, the Netherlands and Flanders samples in our study [1,56]. Even between very similar societies, differences were found with the UK sample, warranting separate norms for the Netherlands [4]. It is clear that motor development and competency depend on many factors (SES, cultural beliefs, exposure, availability). Moreover, the association between SES and functional motor-skill levels seems to be culturally dependent; children from lower SES levels may participate more in active transportation and develop better locomotor skills but may have less PE, less well-equipped sports facilities and less formal sports participation, and thus be less skilled in sport-related movement skills. In developed countries, motor competence is related to SES starting from preschool age [20,48,49,64], while in developing countries, such as South Africa, results are diverging and sometimes in favor of children with low SES [65,66,67]. 

### Limitations of the Study

Many factors known to influence motor development have not been reviewed for this study because data were anonymized and only country, gender and age were available for all children in the sample.

## 5. Conclusions

Our results show that when researchers and practitioners choose a measurement instrument to evaluate motor performance, it is important that they consider possible cultural and gender bias of the items included in that test to measure the ability to perform a wide range of motor skills, as well as the cultural background of the normative sample to which they are comparing the tested children. Clinicians should also be aware that the MABC-2 contains more items that focus on assessing motor skills in which girls tend to be superior, and as such may lead to overrepresentation of boys with lower motor competency even when separate norms are available.

## Figures and Tables

**Figure 1 children-10-01159-f001:**
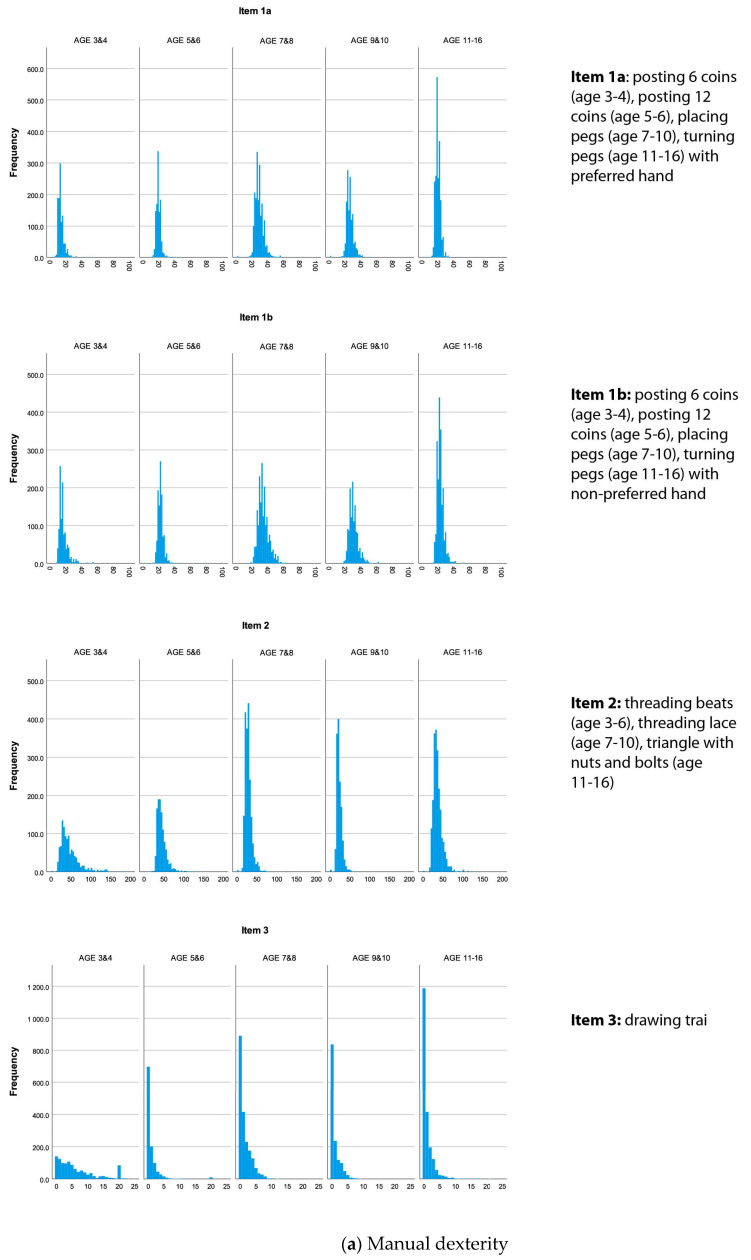
(**a**) Frequency histograms of each MABC-2 manual dexterity item. (**b**) Frequency histograms of each MABC-2 aiming and catching item. (**c**) Frequency histograms of each MABC-2 balance item.

**Table 1 children-10-01159-t001:** Overview of the significant differences between boys and girls based on normative samples.

Authors	Origin of the Sample (Country)	Age Range of the Sample (Years)		Subscales			Total Score
				Manual Dexterity	Aiming and Catching	Balance	
Amador-Ruiz et al., 2018 [41]	Spain	4–5	Item level, raw scores	girls = boys	girls = boys	girls = boys	NR
		6		girls = boys	girls < boys	girls > boys	
Fairbairn et al., 2020 [39]	Australia	8–9	Domain and total standard scores	girls > boys	girls = boys	girls = boys	girls > boys
Hirata et al., 2018 [11]	Japan	3–6	Domain and total standard scores	girls = boys	girls = boys	girls > boys	girls > boys
Kita et al., 2016 [12]	Japan	7–10	Domain and total standard scores	girls > boys	girls = boys	girls > boys	girls > boys
Koksteijn et al., 2018 [7]	Czechia	3–6	Domain and total percentile scores	girls > boys	girls = boys	girls > boys	girls > boys
		3		girls > boys	girls = boys	girls > boys	girls > boys
		4		girls > boys	girls = boys	girls > boys	girls > boys
		5		girls = boys	girls = boys	girls = boys	girls = boys
		6		girls = boys	girls < boys	girls = boys	girls = boys
Meciás-Calvo et al., 2021 [40]	Spain	4–5 *	Domain and total standard scores	girls > boys	girls < boys ^$^	girls > boys	girls > boys
Navarro-Patón et al., 2021 [19]	Spain	5 *	Domain and total standard and percentile scores	girls > boys	girls = boys	girls > boys	girls > boys
Olesen 2014 [42]	Denmark	5–6	Domain standard scores	girls = boys	girls < boys	girls > boys	NR

Legend: NR: Not reported; * children were included for the analyses if their total MABC-2 score was not below the 5th percentile; ^$^ only for children in private schools; no differences were found between boys and girls in public schools. 
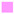
 Girls superior 
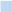
 Boys superior 
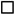
 Similar results for boys and girls.

**Table 2 children-10-01159-t002:** Description of the sample.

		European	African	Total	Subsample Differences
Sample size (boys) (*n*)		5757 (2888)	1897 (937)	7654 (3825)	*p* = 0.560 *
Age band (boys) (*n*)	1	2090 (1087)	114 (62)	2204 (1149)	*p* < 0.001 *
	2	2345 (1183)	1037 (518)	3382 (1701)	
	3	1322 (618)	746 (357)	2068 (975)	
Age (years; mean (SD))		8.1 (3.5)	10.0 (2.8)	8.6 (3.4)	*p* < 0.001 ^§^
Age 3 (*n*)		615	0	615	
Age 4 (*n*)		476	0	476	
Age 5 (*n*)		476	0	476	
Age 6 (*n*)		522	114	636	
Age 7 (*n*)		672	272	944	
Age 8 (*n*)		724	329	1053	
Age 9 (*n*)		533	305	838	
Age 10 (*n*)		417	131	548	
Age 11 (*n*)		286	155	441	
Age 12 (*n*)		300	167	467	
Age 13 (*n*)		220	157	377	
Age 14 (*n*)		164	122	286	
Age 15 (*n*)		161	73	234	
Age 16 (*n*)		191	72	263	

Legend: SD: standard deviation * Chi-squared test, ^§^ Independent sample *t*-test.

**Table 3 children-10-01159-t003:** Item performance for boys and girls across the entire sample (quartiles and *p*-values).

Age Band (*n*)	Domain	Items	Outcome	Boys			Girls			Mann-Whitney U Test
				P25	P50	P75	P25	P50	P75	*p*-Value
1 (2204)	Manual dexterity	1a: Posting coins (preferred)	Time	18	15	11	18	15	11	0.374
		1b: Posting coins (non-preferred)	Time	21	18	14	21	18	14	0.627
		2: Threading beads	Time	54	43	35	48	39	32	<0.001
		3: Drawing trail	Errors	5	2	0	4	1	0	<0.001
	Aiming and catching	4: Catching bean bag	Count	5	8	9	5	7	9	0.025
		5: Throwing bean bag	Count	3	5	7	3	5	7	<0.001
	Balance	6a: One-leg balance (preferred)	Time	6	16	30	7	21	30	<0.001
		6b: One-leg balance (non-preferred)	Time	3	6	17	3	10	23	<0.001
		7: Walking heels raised	Count	6	13	15	8	15	15	<0.001
		8: Jumping on mats	Count	4	5	5	5	5	5	<0.001
2 (3382)	Manual dexterity	1a: Placing pegs (preferred)	Time	30	27	24	28	25	23	<0.001
		1b: Placing pegs (non-preferred)	Time	35	31	28	34	30	26	<0.001
		2: Threading lace	Time	31	26	22	28	24	20	<0.001
		3: Drawing trail	Errors	2	1	0	2	0	0	<0.001
	Aiming and catching	4: Catching ball	Count	6	8	9	4	7	8	<0.001
		5: Throwing bean bag onto mat	Count	5	7	8	5	6	8	<0.001
	Balance	6a: One-board balance (preferred)	Time	12	22	30	16	30	30	<0.001
		6b: One-board balance (non-preferred)	Time	6	12	25	8	17	30	<0.001
		7: Walking heel-to-toe forward	Count	15	15	15	15	15	15	<0.001
		8a: Hopping on mats (preferred)	Count	5	5	5	5	5	5	<0.001
		8b: Hopping on mats (non-preferred)	Count	4	5	5	5	5	5	<0.001
3 (2068)	Manual dexterity	1a: Turning pegs (preferred)	Time	21	18	17	20	18	16	<0.001
		1b: Turning pegs (non-preferred)	Time	24	21	19	24	21	19	0.222
		2: Triangle with nuts and bolts	Time	42	34	29	42	34	28	0.956
		3: Drawing trail	Errors	2	0	0	1	0	0	<0.001
	Aiming and catching	4a: Catching (preferred)	Count	8	10	10	5	8	9	<0.001
		4b: Catching (non-preferred)	Count	7	8	10	3	6	8	<0.001
		5: Throwing ball at wall-mounted target	Count	5	6	8	4	5	7	<0.001
	Balance	6: Two-board balance	Time	13	28	30	13	28	30	0.753
		7: Walking toe-to-heel backwards	Count	11	15	15	9	15	15	0.013
		8a: Zig-zag hopping (preferred)	Count	5	5	5	5	5	5	0.461
		8b: Zig-zag hopping (non-preferred)	Count	5	5	5	5	5	5	0.190

**Table 4 children-10-01159-t004:** Summary of the sex differences across different subsamples.

		European Sample			African Sample		
Domain	Items	AB 1 (Age 6)	AB 2 (Age 7–10)	AB 3 (Age 11–16)	AB 1 (Age 6)	AB 2 (Age 1–10)	AB 3 (Age 11–16)
Manual dexterity	Item 1a						
	Item 1b						
	Item 2						
	Item 3						
Aiming and catching	Item 4a						
	Item 4b						
	Item 5						
Balance	Item 6a						
	Item 6b						
	Item 7						
	Item 8a						
	Item 8b						

Legend: AB1: age band 1; AB2: age band 2; AB3: age band 3. 
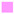
 Girls superior 
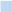
 Boys superior 
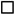
 Similar results for boys and girls 
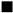
 Not applicable to the age band.

## Data Availability

The data were collected in previous studies commissioned by Pearson and remain their property. As such, the data cannot be made available by the authors of this paper.

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
