# Peer review of "Do Girls Have an Advantage Compared to Boys When Their Motor Skills Are Tested Using the Movement Assessment Battery for Children, 2nd Edition?"

_children, 2023, doi:10.3390/children10071159_

Round 1

Reviewer 1 Report

Title: Do girls have an advantage compared to boys when their motor skills are tested using the Movement Assessment Battery for Children second edition?

Article Type: original scientific paper

Summary

This study has compared AMBC-2 item scores between two groups of girls and boys. Data of 7654 children with a mean age of 8.6 was analyzed for both girls and boys as well as different origin of the sample. The results indicated there are sex-related differences in all age bands. The results also showed that in AB1 and AB2, most manual dexterity and balance items but not aiming and catching items were in favor of the girls. In summary, the results of this study indicated that separate norms for boys and girls are needed.

Evaluation

The topic of this study was interesting for me and even for publication in the Journal. The design for the study was appropriate to answer the research questions. For an original scientific paper, the manuscript was quite straightforward.

Minor points and suggestions

Please add an introduction to the abstract.

Please write a little about sex difference of motor competence in regard of the other related tests such as lincoln-oseretsky test and or TGMD-3 in introduction.

 Please write more about limitation the study in discussion section.

How did you calculate your sample size? How did you consider difference age of children?

Author Response

REVIEWER 1

Summary: This study has compared AMBC-2 item scores between two groups of girls and boys. Data of 7654 children with a mean age of 8.6 was analyzed for both girls and boys as well as different origin of the sample. The results indicated there are sex-related differences in all age bands. The results also showed that in AB1 and AB2, most manual dexterity and balance items but not aiming and catching items were in favor of the girls. In summary, the results of this study indicated that separate norms for boys and girls are needed.

Evaluation: The topic of this study was interesting for me and even for publication in the Journal. The design for the study was appropriate to answer the research questions. For an original scientific paper, the manuscript was quite straightforward.

Minor points and suggestions

  1. Please add an introduction to the abstract.

REPLY: Thank you for your suggestion. The word limit for the abstract is 200 words and our abstract now contains 189 words. Unfortunately, with only 11 words left, we cannot add any additional background. We did follow the journal’s instructions: “The abstract should be a total of about 200 words maximum. The abstract should be a single paragraph and should follow the style of structured abstracts, but without headings: 1) Background: Place the question addressed in a broad context and highlight the purpose of the study; 2) Methods: Describe briefly the main methods or treatments applied. Include any relevant preregistration numbers, and species and strains of any animals used; 3) Results: Summarize the article's main findings; and 4) Conclusion: Indicate the main conclusions”. We hope the reviewer understands why we could not meet their recommendation.

  1. Please write a little about sex difference of motor competence in regard of the other related tests such as lincoln-oseretsky test and or TGMD-3 in introduction.

REPLY: We agree that sex differences are present in other tests and added the following sentences:

p.2, lines 78-88: “This is surprising, since it is well-established that boys outperform girls on object control tasks [13-16], and girls are better at locomotor [16-18] and balance tasks [13-15]. These differences are usually assigned to how they spend their leisure time: boys usually play more with a ball, and girls do activities such as rope skipping and other balancing tasks related to gymnastics [19]. Several standardized motor tests do acknowledge sex differences – as mentioned above – through their separate norms for boys and girls, e.g. the Bruininks-Oseretsky Test of Motor Proficiency, 2nd edition (BOT-2; boys better at strength, running speed and agility tasks) [20], the Test of Gross Motor Development, 2nd and 3rd edition (boys better at object control and girls at locomotor tasks [21]), the Körperkoordinationstest für Kinder 3+ (boys better at side jumping and eye-hand coordination, girls better at backward walking) [19].”

p.3, lines 109-117: “For instance, even though, worldwide, boys are better at ball skills, even as young as 15-23 months old [45], Aboriginal girls throw better than children from other cultures, which can be explained by their cultural belief that throwing for hunting and defense is important for both sexes [46]. Another aspect that that has been stipulated to impact motor skill development is the socio-economic status (SES). Several authors indicate that low SES has a negative impact on motor skills in children [18, 47, 48], whereas others do not report differences [49]. Whether or not SES has a negative impact on the SES seems to depend upon the region and the type of skills being assessed [13, 50-52], but the interaction effect with sex remains under debate [48, 52].”

  1. Please write more about limitation the study in discussion section.

REPLY: Thank you for your suggestions. We added the following sentence to the discussion section (p. 9, lines 336-339): “Limitations of the study: Many factors known to influence motor development have not been reviewed for this study because data were anonymized and only country, gender and age were available for all children in this sample.”

  1. How did you calculate your sample size? How did you consider difference age of children?

REPLY: In this paper we combined previously collected data and performed a secondary analysis, which is why we did not perform an a priori sample size calculation. The norms for the MABC-2 were established for three- to 16-year-old children, because this was the intent of the test developers. The African sample was one of convenience, where the MABC-2 was administered as a reference test to validate other motor scales, intended for use in low-resourced areas.

Reviewer 2 Report

The aim of the study was to investigate gender-related differences in raw item scores of the Movement Assessment Battery for Children, 2nd edition (MABC-2) in a large dataset collected in different regions around the world, and to try to clarify whether there is an interaction effect between gender and sample origin (Europeans versus Africans). I would like to congratulate and thank them for their effort and motivation involved in this research study. The presentation of the research is well documented, with a scientific basis and respects the latest standards regarding the highest level scientific publications. The methodology was chosen correctly. The conclusions support and result from the research and open new directions for future research. The submitted work is interesting and essentially exhausts the subject under discussion. I have only a few minor suggestions: What the research hypotheses for the study were? Did the study confirm them? Were these results expected? This should be further elaborated at the end of the introduction. Furthermore, I do not see any study limitations in the article. Please complete it with at least one comprehensive discussion paragraph or create a separate paragraph before the conclusions. Supplementing the article with the above-mentioned scope will in my opinion make a real chance for publication in Children. I keep my fingers crossed for the final success of the publication. 

Author Response

REVIEWER 2

Summary: The aim of the study was to investigate gender-related differences in raw item scores of the Movement Assessment Battery for Children, 2nd edition (MABC-2) in a large dataset collected in different regions around the world, and to try to clarify whether there is an interaction effect between gender and sample origin (Europeans versus Africans). I would like to congratulate and thank them for their effort and motivation involved in this research study.

Evaluation: The presentation of the research is well documented, with a scientific basis and respects the latest standards regarding the highest-level scientific publications. The methodology was chosen correctly. The conclusions support and result from the research and open new directions for future research. The submitted work is interesting and essentially exhausts the subject under discussion.

I have only a few minor suggestions:

  1. What the research hypotheses for the study were? Did the study confirm them? Were these results expected? This should be further elaborated at the end of the introduction.

REPLY: Thank you for your suggestions. We added a few sentences about the hypotheses in the introduction section (p. 3, lines 130-135): “Since the domain scores are the scaled sum of the standardized item scores, sex-related differences at item level should be explored first. We hypothesize that overall girls will outperform boys on the manual dexterity and balance items, and that boys will be better at aiming and catching than girls. Because of the daily activities of the children and their sports participation, we expect that these differences will be clearly present in the European sample.”.

  1. Furthermore, I do not see any study limitations in the article. Please complete it with at least one comprehensive discussion paragraph or create a separate paragraph before the conclusions. 

REPLY: Thank you for your suggestion. We added the following sentence to the discussion section (p. 9, lines 336-339): “Limitations of the study: Many factors known to influence motor development have not been reviewed for this study because data were anonymized and only country, gender and age were available for all children in this sample.”

  1. Supplementing the article with the above-mentioned scope will in my opinion make a real chance for publication in Children. I keep my fingers crossed for the final success of the publication.

REPLY: Thank you.